# Estimating Mutual Information for Spike Trains: A Bird Song Example

**DOI:** 10.3390/e25101413

**Published:** 2023-10-03

**Authors:** Jake Witter, Conor Houghton

**Affiliations:** Faculty of Engineering, University of Bristol, Bristol BS8 1TR, UK; conor.houghton@bristol.ac.uk

**Keywords:** mutual information, spike trains, zebra finch, spike train metric

## Abstract

Zebra finches are a model animal used in the study of audition. They are adept at recognizing zebra finch songs, and the neural pathway involved in song recognition is well studied. Here, this example is used to illustrate the estimation of mutual information between stimuli and responses using a Kozachenko–Leonenko estimator. The challenge in calculating mutual information for spike trains is that there are no obvious coordinates for the data. The Kozachenko–Leonenko estimator does not require coordinates; it relies only on the distance between data points. In the case of bird songs, estimating the mutual information demonstrates that the information content of spiking does not diminish as the song progresses.

## 1. Introduction

The mutual information between two random variables *X* and *Y* is often conveniently described using a diagram like this:

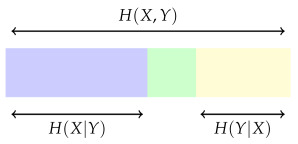

where the whole rectangle represents the entropy H(X,Y) of the joint variable (X,Y). This is, in general, less than the sum of H(X) and H(Y) because *X* and *Y* are not independent. In this diagram, the purple and green regions together are intended to represent H(X), and the green and yellow regions are intended to represent H(Y). The purple region on its own represents H(X|Y): the entropy remaining, on average, when the value of *Y* is known. In the same way, the yellow region represents H(Y|X). Now, the mutual information is represented by the green section:



It is
(1)I(X,Y)=H(X)−H(X|Y)=H(Y)−H(Y|X)
or, by substitution,
(2)$$I\left(X,Y\right)=E\log_{2}\left[\frac{p_{X|Y}\left(x|y\right)}{p_{X}\left(x\right)}\right]=E\log_{2}\left[\frac{p_{Y|X}\left(y|x\right)}{p_{Y}\left(y\right)}\right]$$

Here, for illustrative purposes, mutual information is described relative to a specific example: the neural response of cells in the zebra finch auditory pathway to zebra finch songs. This is both an interesting neuroscientific example and an example which is typical of a broad set of neuroscience problems.

The zebra finch is a model animal used to study both auditory processing and learning; the male finch sings, and he has a single song which begins with a series of introductory notes, followed by two or three repetitions of the motif: a series of complex frequency stacks known as syllables, separated by pauses. Syllables are about 50 ms long, with songs lasting about two seconds. The songs have a very rich structure, and both male and female zebra finches can distinguish one zebra finch song from another.

Here, we use a data set consisting of spike trains recorded while the bird is listening to a set of songs, and we provide an estimate for the mutual information between the song identity and spike trains recorded from cells in the auditory pathway. This is an interesting and non-trivial problem. Generally, calculating mutual information is costly in terms of data because it requires the estimation of probabilities such as $p_{Y}\left(y\right)$ and $p_{Y|X}\left(y|x\right)$. For this reason, some measure of correlation is often used when quantifying the relationship between two random variables. However, not all data types have a correlation: calculating the correlation assumes algebraic properties of the data that are not universal. As an example, calculating the correlation between *X* and *Y* requires the calculation of E[XY], which in turn assumes that it makes sense to multiply the *x* and *y* values. This is not the case for the typical neuroscience example considered here, where the set of outcomes for *X* is the song identities and that for *Y* is the spike trains. To circumvent this, spike trains are often replaced with something else, spike counts for example. However, this involves an implicit assumption about how information is coded. This is likely to be inappropriate in many cases. Indeed, the approach taken to calculating mutual information can involve making very strong assumptions about information coding, the very thing that is being studied.

The purpose of this review paper is to demonstrate a different approach: there is a metric-space version of the Kozachenko–Leonenko estimator [1,2] introduced in [3,4,5] and inspired by [6]. This approach has been tested on simulated data, for example in [5], and this shows it to be promising. However, it is important to also test it on real data. Here, it is applied in the zebra finch example.

## 2. Materials and Methods

Let
(3)$$D=\{\left(x_{1},y_{1}\right),\left(x_{2},y_{2}\right),\;...\;,\left(x_{n},y_{n}\right)\}$$
be a data set. In our case, the $x_{i}$ are the labels for songs in the set of stimuli, with each $x_{i}\in \left\{1,\ldots ,n_{s}\right\}$; $n_{s}$ is the number of different songs. For a given trial, $y_{i}$ is the spiking response. This will be a point in “the space of spike trains”. What exactly is meant by the space of spike trains is less clear, but for our purposes here, the important point is that this can be regarded as a metric space, with a metric that gives a distance between any two spike trains; see [7,8], or, for a review, [9].

Given the data, the mutual information is estimated by
(4)$$I\left(X,Y\right)\approx \frac{1}{n}\sum_{i=1}^{n}\log_{2}\left[\frac{p_{Y|X}\left(y_{i}|x_{i}\right)}{p_{Y}\left(y_{i}\right)}\right]$$
where the particular choice of which conditional probability to use, $p_{Y|X}$ rather than $p_{X|Y}$, has been made for later convenience. Thus, the problem of estimating mutual information is one of estimating the probability mass functions $p_{Y|X}$ and $p_{Y}$ at the data points in D. In our example, there is no challenge to estimating $p_{X}$, since each song is presented an equal number of times during the experiment $p_{X}\left(x_{i}\right)=1/n_{s}$ for all $x_{i}$ and, in general $p_{X}\left(x_{i}\right)$ is known from the experiment design. However, estimating $p_{Y|X}$ and $p_{Y}$ is more difficult.

In a Kozachenko–Leonenko approach, this is performed by first noting that for a small volume $R_{i}$ containing the point $y_{i}$,
(5)$$p_{Y}\left(y_{i}\right)\approx \frac{1}{\mathrm{vol}(R_{i})}\int_{R_{i}}p_{Y}\left(y\right)\;dy$$
with the estimate becoming more and more exact for smaller regions $R_{i}$. If the volume of $R_{i}$ were reduced towards zero, $p_{Y}\left(y\right)$ would be constant in the resulting tiny region. Here, $\mathrm{vol}(R_{i})$ denotes the volume of $R_{i}$. Now, the integral $\int_{R_{i}}p_{Y}\left(y\right)\;dy$ is just the probability mass contained in $R_{i}$ and so it is approximated by the number of points in D that are in $R_{i}$:(6)$$\int_{R_{i}}p_{Y}\left(y\right)\;dy\approx \frac{\left|\{y_{j}\in R_{i}\}\right|}{n}.$$

It should be noted at this point that this approximation becomes more and more exact as $R_{i}$ becomes bigger. Using the notation
(7)$$k_{i}=\left|\left\{y_{j}\in R_{i}\right\}\right|$$
this means
(8)$$p_{Y}\left(y_{i}\right)\approx \frac{k_{i}}{n\mathrm{vol}(R_{i})}.$$

This formula provides an estimate for $p_{Y}\left(y_{i}\right)$ provided that a strategy is given for choosing the small regions $R_{i}$ around each point $y_{i}$. As will be seen, a similar formula can be derived for $p_{Y|X}\left(y_{i}|x_{i}\right)$, essentially by restricting the points to $D_{i}=\left\{\left(x_{j},y_{j}\right)\in D|x_{j}=x_{i}\right\}$:(9)$$p_{Y|X}\left(y_{i}|x_{i}\right)\approx \frac{h_{i}}{n_{c}\mathrm{vol}\left(R_{i}\right)}$$
where $h_{i}$ is the number of points in $R_{i}$ with label $x_{i}$ and $n_{c}$ is the total number of points with label $x_{i}$. In the example here, $n_{c}=n/n_{s}$. Once the probability mass functions are estimated, it is easy to estimate the mutual information. However, there is a problem: the estimates also require the volume of $R_{i}$. In general, a metric space does not have a volume measure. Furthermore while many everyday metric spaces also have coordinates providing a volume measure, this measure it not always appropriate since the coordinates are not related to the way the data are distributed. However, the space that the $y_{i}$’s belong to is not simply a metric space, it is also a space with a probability density, $p_{Y}\left(y\right)$. This provides a measure of volume:(10)$$\mathrm{vol}\left(R_{i}\right)=\int_{R_{i}}p_{Y}\left(y\right)dy$$

In short, the volume of a region can be measured as the amount of probability mass it contains. This is useful because this quantity can in turn be estimated from data, as before, by counting points:(11)$$\mathrm{vol}\left(R_{i}\right)\approx \frac{k_{i}}{n}.$$

The problem with this, though, is that it gives a trivial estimate of the probability. Substituting back into the estimate for $p_{Y}\left(y_{i}\right)$, Equation (Equation 8) gives $p_{Y}\left(y_{i}\right)=1$ for all points $y_{i}$. This is not as surprising as it might at first seem. Probability density is a volume-measure dependent quantity; that is what is meant by calling it a density and is the reason that entropy is not well defined on continuous spaces. There is always a choice of coordinate that trivializes the density.

However, it is not the entropy that is being estimated here. It is the mutual information and this is well defined: its value does not change when the volume measure is changed. The mutual information uses more than one of the probability densities on the space; in addition to $p_{Y}\left(y_{i}\right)$, it involves the conditional probabilities $p_{Y|X}\left(y|x\right)$. Using the measure defined by $p_{Y}\left(y\right)$ does not make these conditional probability densities trivial. The idea behind the metric space estimator is to use $p_{Y}\left(y\right)$ to estimate volumes. This trivializes the estimates for $p_{Y}\left(y_{i}\right)$, but it does allow us to estimate $p_{Y|X}\left(y|x\right)$ and use this to calculate an estimate of the mutual information.

In this way, the volume of $R_{i}$ is estimated from the probability that a data point is in $R_{i}$, and this, in turn, is estimated by counting points. Thus, to fix the volume $\mathrm{vol}(R_{i})$, a number *h* of data points is specified, and for each point, the h−1 nearest data points are identified, giving *h* points in all when the “seed point” is included. This is equivalent to expanding a ball around $y_{i}$ until it has an estimated volume of h/n. This defines the small region $R_{i}$. The conditional probability is then estimated by counting how many points in $R_{i}$ are points with label $x_{i}$, that is, are points in $D_{i}$. In fact, this just means counting how many of the *h* points that have been identified are in $D_{i}$, or, put another way, it means counting how many of the h−1 nearest points to the original seed point are from the same stimulus as the seed point. In summary, the small region consists of *h* points. To estimate $p_{Y|X}\left(y_{i}|x_{i}\right)$, the number of points in the small region corresponding to label $x_{i}$ is counted; this is referred to as $h_{i}$ so
(12)$$h_{i}=|\left\{y_{j}\in R_{i}\;|\;x_{j}=x_{i}\right\}\left|=\right|R_{i}\cap D_{i}|.$$

This is substituted into the formula for the density estimator, Equation (Equation 6), to obtain
(13)$$p_{Y|X}\left(y_{i}|x_{i}\right)\approx \frac{n}{n_{c}}\frac{h_{i}}{h}$$
where, as before, $n_{c}$ is the total number of trials for each song. It is assumed that each song is presented the same number of times. It would be easy to change this to allow for different numbers of trials for each song, but this assumption is maintained here for notational convenience. Substituting back into the formula for the estimated mutual information, Equation (Equation 4) gives
(14)$$I_{0}=\frac{1}{n}\sum_{i=1}^{n}\log_{2}\frac{n_{s}h_{i}}{h}$$

The calculation of $I_{0}$ is illustrated in Figure 1. The subscript zero has been added in order to preserve the unadorned *I* for the information itself and $\tilde{I}$ for the de-biased version of the estimator; this is discussed below.

This estimate is biased, and it gives a non-zero value even if *X* and *Y* are independent. This is a common problem with estimators of mutual information. One advantage of the Kozachenko–Leonenko estimator described here is that the bias at zero mutual information can be calculated exactly. Basically, for the estimator to give a value of zero $h_{i}=h/n_{s}$ would be required for every *i*. In fact, while this is the expected value if *X* and *Y* are independent, $h_{i}$ has a probability distribution which can be calculated as a sort of an urn problem. As detailed in [10], performing this calculation gives the de-biased estimator:(15)$$I\approx \tilde{I}=I_{0}-I_{b}$$
where $I_{b}$, the bias, is
(16)$$I_{b}=\sum_{r=1}^{h}\sum_{c=1}^{n_{s}}\frac{n_{c}}{n}u\left(r-1;n_{c}-1,h-1,n-n_{c}\right)\log_{2}\frac{n_{c}r}{h}$$
and *u* is the probability for the hypergeometric distribution using the parameterization used by the distributions.jl Julia library.
(17)u(k;s,h,f)=skfm−ks+fm≡Hypergeometric(s,m,f)

Obviously, the estimator relies on the choice of smoothing parameter *h*. Recall that for a small *h*, the counting estimates for the number of points in the small region and for the volume of the small regions are noisy. For a large *h*, the assumption that the probability density is constant in the small region is poor. These two countervailing points of approximation affect $I_{0}$ and $I_{b}$ differently. It seems that a good strategy in picking *h* for real data is to maximize $\tilde{I}\left(h\right)$ over *h*. This is the approach that will be adopted here.

As an example, we will use a data set recorded from zebra finches and made available on the Collaborative Research in Computational Neuroscience data sharing website [11]. This data set contains a large number of recordings from neurons in different parts of the zebra finch auditory pathway. The original analysis of these data are described in [12,13]. The data set includes different auditory stimuli; here, though only the responses to zebra finch song are considered. There are 20 songs, so $n_{s}=20$, and each song is presented ten times, $n_{c}=10$, giving n=200. The zebra finch auditory pathway is complex and certainly does not follow a single track, but for our purposes, it looks like
(18)auditorynerve→CN→MLd→OV→FieldL→HVc
where CN is the cochlear nuclei; MLd is the mesencephalicus lateralis pars dorsalis, analogous to mammalian inferior colliculus; OV is the nucleus ovoidalis; Field L is the primary auditory pallium, analogous to mammalian A1; and, finally, HVc is regarded as the locus of song recognition. The mapping of the auditory pathway and our current understanding of how to best associate features of this pathway to features of the mammalian brain is derived from, for example [13,14,15,16,17].

In the data set, there are 49 cells from each of MLd and Field L, and here, the entropy is calculated for all 98 of these cells.

## 3. Results

Our interest in considering the mutual information for bird songs was to check whether the early part of the spike train was more informative about the song identity. It seemed possible that the amount of information later in the spike train would be less than in the earlier portion. This does not seem to be the case.

There are a number of spike train metrics than could be used. Although these differ markedly in the mechanics of how they calculate a distance, it does appear that the more successful among them are equally good at capturing the information content. In Figure 2A, the total mutual information between song identity and spike train is plotted. Here, the Victor–Purpura (VP) metric [7], the spike count, the Earth mover distance (EMD) [18], and the van Rossum metric [8] are considered. The Victor–Purpura metric and van Rossum metric both include a parameter which can be tuned, roughly corresponding to the precision of spike timing. Here, the optimal value for each case has been used, chosen to maximize the average information. These values are q=32.5 Hz for the VP metric and τ=15 ms for the vR metric. The mutual information estimator uses the metric to order the points, and each small region contains the h−1 points nearest the seed point so the estimator does not depend on the distances themselves, just the order. Indeed, the estimated mutual information is not very sensitive to the choice of *q* or τ. This is demonstrated in Figure 2B, where the mutual information is calculated as a function of *q*, the parameter for the VP metric.

The Victor–Purpura metric and van Rossum metric clearly have the highest mutual information and are very similar to each other. This indicates that the estimator is not sensitive to the choice of metric, provided the metric is one that can capture features of the spike timing as well as the overall rate. The spike count does a poor job, again indicating that there is information contained in spike timing as well as the firing rate. Similar results were seen in [9,19], though a different approach to evaluating the performance of the metrics was used there.

The cells from MLd have higher mutual information, on average, than the cells from Field L. Since Field L is further removed from the auditory nerve than MLd, this is to be expected from the information processing inequality. This inequality stipulates that away from the source of information, information can only be lost, not created.

In Figure 3, the information content of the spike trains as a function of time is considered. To achieve this, the spike trains are sliced into 100 ms slices and the information is calculated for each slice. The songs have variable lengths, so the mutual information becomes harder to interpret after the end of the shortest song, marked by a dashed line. Nonetheless, it is clear that the rate of information and the information per spike are largely unchanged through the song.

## 4. Discussion

As well as demonstrating the use of the estimator for mutual information, we were motivated here by an interest in the nature of coding in spike trains in a sensory pathway. It is clear that the neurons in MLd and Field L are not “grandmother” neurons, responding only to a specific song and only through the overall firing rate. The firing rate contains considerably less information than was measured using the spike metrics. The spike metrics, in turn, give very similar values for the mutual information; this appears to indicate that the crucial requirement of a spike train metric is a “fuzzy” sensitivity to spike timing. This demonstrates the need for an estimator such as the Kozachenko Leonenko estimator used here.

Approaches that do not incorporate spike timings underestimate the mutual information, but histogram methods, which do include timings, are computational impractical for modest amounts of data. A pioneering paper, [19], also examines mutual information for zebra finch songs, but using a histogram approach. The substantial conclusion there was similar to the conclusion here: there was evidence that spike timings are important. However, it seems likely that this early paper was constrained in its estimates by the size of the data set. This is suggested by the way the amount of information measured increased monotonically as the bin-width in the temporal discretization was reduced, a signature of a data-constrained estimate.

Finally, it is observed that it is not the case that the precision of spiking diminishes as the song continues. Since that song can often be identified from the first few spikes of the response, it might be expected that the neuronal firing would become less precise. Precision is metabolically costly. However, although the firing rate falls slightly, the information remains constant on a per-spike basis.

## Figures and Tables

**Figure 1 entropy-25-01413-f001:**
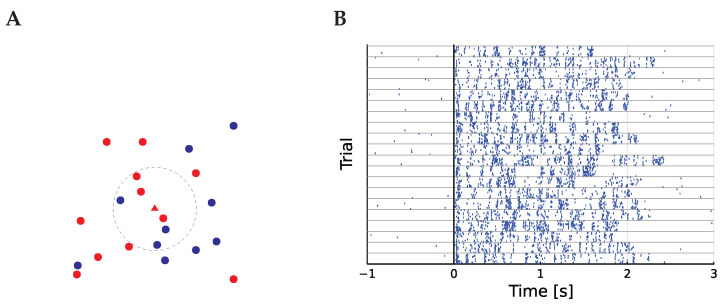
The calculation of *I* and the spiking data. (**A**) illustrates how the estimator is calculated. The circles and triangle are data points, and red and blue represent two different labels. The dashed line is the small region around the seed point in the center marked by a triangle ▲ while the other, non-seed points are circles: ● and ●. Here, h=7, so the ball has been expanded until it includes seven points. It contains four red points, the colour of the central point, so $h_{▲}=4$. For illustration, the points have been drawn in a two-dimensional space, but this can be any metric space. (**B**) describes the data. The spiking responses of a typical neuron to each presentation of a song is plotted as a raster plot, with a mark for each spike. The trials are grouped by song, so the ten responses in each group correspond to repeated presentations of a single stimulus. Stimulus onset is aligned at 0, with the shortest song lasting 1.65 s.

**Figure 2 entropy-25-01413-f002:**
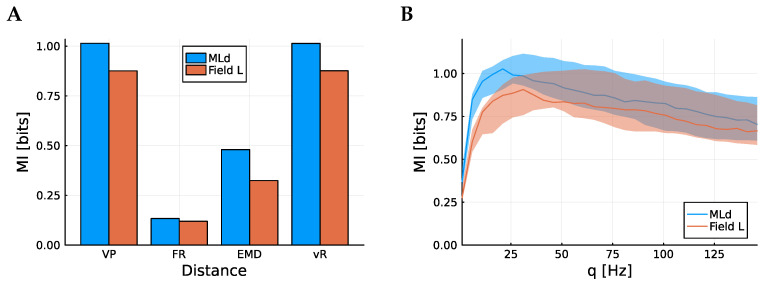
Information content according to different distances. (**A**) shows mean mutual information (MI) among the 98 neurons from both regions according to different distance metrics, the Victor–Purpura metric, the firing rate, the Earth mover distance, and the van Rossum metric. To calculate the mutual information, 1.65 s of spike train is used, corresponding to the length of the short song. (**B**) shows how that mean MI varies according to the *q* parameter for the Victor–Purpura metric. In both cases, blue corresponds to MLd and red corresponds to Field L. In (**B**), the translucent band corresponds to the middle 20% of data points; there is substantial variability in information across cells.

**Figure 3 entropy-25-01413-f003:**
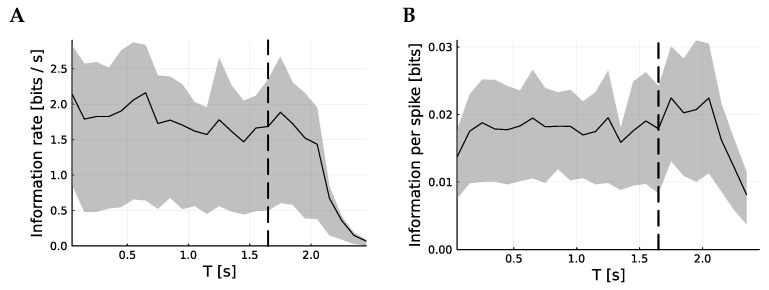
Information content per time. These figures show the time-resolved mutual information by calculating the mutual information for spiking response over 0.1 s slices; the centers of which, *T*, are plotted against the mean mutual information. (**A**) shows how this varies over time, with a vertical line showing the ending of the shortest stimulus. (**B**) shows the mean information per spike; although (**A**) shows a small decrease, (**B**) seems to indicate that this corresponds to a reduction in firing rate, not in the information contained in each spike. In both cases, the metric is the VP metric with q=30 Hz.

## Data Availability

Not applicable.

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
