# Peer review of "Estimating Mutual Information for Spike Trains: A Bird Song Example"

_entropy, 2023, doi:10.3390/e25101413_

Round 1
Reviewer 1 Report
On the example of Zebra finch the Authors illustrate an interesting estimation method of mutual information between stimulus and response using a Kozachenko-Leonenko estimator. The challenge in calculating mutual information for spike trains is that there are no obvious coordinates for the data.
I have some remark/questions
1. Page 2, in (4) put log_2 instead of log
2. Page 3, line 88, put for instead of fpr
3. In calculation (13) you ue P_Y(y_i)=1, point this clearly in text.
4. Page 5 in (15) nx is not defined, it is n_s?
5. In real data analysis how was the smoothing parameter h selected?
6. It would be interesting to check the proposed method for estimating the mutual information on simulation data sets.
7. Page 6, label the colors in Figure 2A.
Author Response
We are grateful to Referee 1 for their overall positive assessment of our work and for the useful comments they have supplied. We are sorry the paper did contain a number of simple errors and are grateful we've had the opportunity to address these. We have followed all of Referee 1's corrections; this is detailed below. We have also attached the resubmitted paper with track changes, this has new text in blue, substantially revised text in teal and deleted text in red.
We hope our paper now merits publication, we are certainly pleased with it!
All the best
Conor
1. Page 2, in (4) put log_2 instead of log
> Thanks for pointing this out, we have fixed this inconsistency.
2. Page 3, line 88, put for instead of fpr
> Thank you. Done.
3. In calculation (13) you use P_Y(y_i)=1, point this clearly in text.
> We have made substantial changes to that relevant paragraph to try to make this clearer.
4. Page 5 in (15) nx is not defined, it is n_s?
> Yes, thank you for spotting that!
5. In real data analysis how was the smoothing parameter h selected?
> This is done by maximizing I-tilde over h; the last paragraph of the "materials and methods" now reads:
"Obviously the estimator relies on the choice of the smoothing parameter $h$; recall that for small $h$ the counting estimates for the number of points in a ball and for the volume of the balls are noisy; for large $h$ the assumption the probability density is constant in the ball is poor. These two countervailing points of approximation affect $I_0$ and $I_b$ differently. It seems a good strategy in picking $h$ for real data is to maximize $\tilde{I}(h)$ over $h$. This is the approach that will be adopted here."
6. It would be interesting to check the proposed method for estimating the mutual information on simulation data sets.
> This has been done in earlier work; one motivation for the current work is that in the past this approach has only been tested on simulated data! This was not at all clear and so we have added:
"This approach has been tested on simulated data, for example in \cite{Houghton2019} and this shows it to be promising. However, it is important to also test it on real data. Here it is applied in the zebra finch example."
to the last paragraph of the introduction.
7. Page 6, label the colors in Figure 2A.
> This has been fixed, thanks!

Reviewer 2 Report
The paper "Estimating mutual information for spike trains: a bird song example" provides a well-reasoned evaluation of the KL estimator for spike streams. While the analysis here is interesting, the paper appears to be written quite casually which becomes a distraction from the overall narrative. For example, even the first sentence of the paper has an inline figure that - while educational - occurs before the reader even knows what the paper is going to be about.
Some specific comments:
There is a lot of variance in the MI estimates across spike train metrics. What is the point of the Kozachenko-Leonenko estimator if it’s not clear which metric we should be using to compute it? Why is the metric that yields the maximum information an appropriate choice? Some more detail connecting the interpretation of each metric with respect to mutual information would be helpful, beyond firing rate/spike timing. As an extreme case, couldn’t we define a spike train metric to evaluate to 0 if the spike trains belong to the same song and nonzero if they belong to different songs?
It would be nice to see the MI computed for spike trains shuffled in time, with respect to the different spike train metrics.
This paper has too many run-on sentences and it hinders understanding the main points of the paper.
There are several irrelevant and distracting sentences, such as “… zebra finch are more prized for their appearance…” (line 32) or (line 142) regarding the meaning of the letters “HVc”. The study doesn’t even look at responses in HVc.
Figure 1B is confusing; I gather these are response for a single cell, but are they arranged so that each of the 20 repetitions of the same song are adjacent? The Y axis is mislabeled, it should be ‘trial’, and it would be helpful if responses from individual songs are labeled.
The demonstration of the Kozachenko-Leonenko estimator on zebra finch spike train data is sound, but the writing makes the presentation difficult to follow. A tremendous amount of wording could be eliminated to improve the flow of the text.
The point made in line 45 is a good point but could be said much more concisely (this sentence has 8 commas).
While this is paper is a helpful explanation of the Kozachenko-Leonenko approach to estimating MI, I don’t think this study adds to our knowledge of the relationship between spiking responses and the acoustic stimuli in songbirds. See for example “Spike timing and the coding of naturalistic sounds in a central auditory area of songbirds”, Wright, Sen, Bialek, Doupe, NIPS 2001. It would be more impactful if this paper contrasted the K-L approach to previous methods that have been used for estimating mutual information in songbird neural responses, and show that it yields some distinct insight.
The paper authors appear proficient in English, however the quality of writing is difficult to parse as a reader. Partly this is style (the introduction comes off like an opening description to a methods section); and partly it is grammar (too many wordy and run-on sentences). I recommend that the authors consider some of the style and tone of other papers in the field (such as the papers cited in this study), and I would recommend that the authors consider getting editorial assistance on the paper
Author Response
We are very grateful to Referee 2 for their thoughtful review and for their guide to better style in presenting this work. We have made a large number of changes to avoid the sort of long sentences and needless information that they criticized, we have also address all their other concerns, a point-by-point responses is below and we have attached a "track-changes" version of the manuscript, in this blue marks new text, teal substantially revised text and red, text that has now been deleted.
We are pleased to have been able to improve our paper in this way and hope the referee agrees that we have benefited from their recommendations.
All the best
Con
There is a lot of variance in the MI estimates across spike train metrics. What is the point of the Kozachenko-Leonenko estimator if it’s not clear which metric we should be using to compute it?
? This is an interesting point, and, indeed, one of the points we wanted to examine in this paper: we would regard this (Figure 2A) as one of the contributions. In fact, it does demonstrate that the result is very consistent when the comparison is made between the VP and vR metrics; it is only the cruder metrics, the FR and EMD metrics that produce a very different answer. In fact, this demonstrates that these metrics do not accurately measure the information, reducing the spike train to its firing rate loses a lot of information. Indeed, Figure 2B demonstrates that mutual information declines gracefully as the value of q is changed.
> The challenge in calculating mutual information is, in a sense, in finding it. Often crude firing rate based information measures are used, we see here that they are dramatically under-estimating the mutual information, so the answer to the question, "why use a KL estimator" is that other approaches are getting the wrong answer!
> We have tried to clarify this by editing the results section to read:
"The Victor-Purpura metric and van Rossum metrics clearly have the highest mutual information and are very similar to each other. This indicates that the estimator is not sensitive to the choice of metric, provided the metric is one that can capture features of the spike timing as well as the overall rate. The spike count does a poor job, again indicating that there is information contained in spike timing as well as the firing rate. Similar results were seen in \cite{HoughtonVictor2010}, albeit with a different choice of metric."
> and by adding the sentence
"This demonstrates the need for an estimator such as the KL estimator used here; approaches that do not incorporate spike timings underestimate the mutual information, but histogram methods, which do include timings are computational impractical."
> to the conclusions.
Why is the metric that yields the maximum information an appropriate choice? Some more detail connecting the interpretation of each metric with respect to mutual information would be helpful, beyond firing rate/spike timing. As an extreme case, couldn’t we define a spike train metric to evaluate to 0 if the spike trains belong to the same song and nonzero if they belong to different songs? It would be nice to see the MI computed for spike trains shuffled in time, with respect to the different spike train metrics.
> We have tried to improve the exposition throughout to clarify this; we did not want to discuss at too great a length the difference between metrics, this has been discussed before, for example in the review we cite by Houghton and Victor. The metric suggested which depends on stimulus would produce maximum entropy, but wouldn't not be measuring an intrinsic property of the spike train, conversely, shuffling the spikes reduces all the measures down to the level of firing rate metrics.
This paper has too many run-on sentences and it hinders understanding the main points of the paper.
> We have split a considerable number of the longer sentences to form shorter sentences which some might find easier to parse.
There are several irrelevant and distracting sentences, such as “… zebra finch are more prized for their appearance…” (line 32) or (line 142) regarding the meaning of the letters “HVc”. The study doesn’t even look at responses in HVc.
>the sentence
"Though derived from an acronym, controversy over what precisely the letters stand for means that ``HVc'' is now regarded as the proper name."
?and
"As a pet animal, zebra finch are more prized for their appearance than for their song"
>have been removed and the description of zebra finch brain anatomy shortened overall.
Figure 1B is confusing; I gather these are response for a single cell, but are they arranged so that each of the 20 repetitions of the same song are adjacent? The Y axis is mislabeled, it should be ‘trial’, and it would be helpful if responses from individual songs are labeled.
>This has been done, the figure is much nicer now, thank you for the help!
The demonstration of the Kozachenko-Leonenko estimator on zebra finch spike train data is sound, but the writing makes the presentation difficult to follow. A tremendous amount of wording could be eliminated to improve the flow of the text.
>we have made numerous changes to improve the flow of the text and have tried to eliminate wording where possible.
The point made in line 45 is a good point but could be said much more concisely (this sentence has 8 commas).
>this has been changed to read:
"To circumvent this, the spike trains are often replaced with something else, spike counts for example. However, this involves an implicit assumption about how information is coded. This assumption is likely to be inappropriate in many cases. Indeed, the approach taken to calculating mutual information can involve making strong assumptions about information coding, the very thing that is being studied."
While this is paper is a helpful explanation of the Kozachenko-Leonenko approach to estimating MI, I don’t think this study adds to our knowledge of the relationship between spiking responses and the acoustic stimuli in songbirds. See for example “Spike timing and the coding of naturalistic sounds in a central auditory area of songbirds”, Wright, Sen, Bialek, Doupe, NIPS 2001. It would be more impactful if this paper contrasted the K-L approach to previous methods that have been used for estimating mutual information in songbird neural responses, and show that it yields some distinct insight.
> We do not wish to be too critical of that paper, it was interesting, pioneering and insightful. However, we believe that the approach is not robust enough for data sets of that size. We have written:
"A pioneering paper \cite{WrightEtAl2001} did examine mutual information for zebra finch song using a histogram approach. The substantial conclusion of this paper was similar to the conclusion here, there was evidence that spike timings are important. However, it is also clear that this early paper suffered from constraints due to the size of the data set. This is clear because the amount of information measured increased monotonically as the bin-width in the temporal discretization was reduced, a signature of a data-constrained estimate."

Reviewer 3 Report
Summary:
The authors examine the problem of estimating the mutual information between bird songs and spike trains. The authors present a useful set of results that are well motivated. However, I have a number of specific comments and suggestions that I believe can improve the clarity and presentation.
Major comments:
There appears to be a discrepancy between Eq. (5) and Eq. (8). Specifically, should V_i be V in the denominator of Eq. (8), or vice versa? This is important because it guides the entire following discussion. I am particularly concerned by the fact that p(y_i) = 1 for all y_i. Should this really be the case?
It is hard to follow the explanation in the paragraph beginning on Line 96. A clearer explanation would help the reader. In particular, the justification for Eq. (11) is not clear. Also, should Eq. (11) read p_{Y|X}(y_i | x_i)?
I am confused by Eq. (13). If p(y_i) = 1, then the only thing inside the log is p(y_i|x_i), which is given in Eq. (11). But this doesn’t match what’s inside the log in Eq. (13).
At the beginning of the Results, can the authors expand upon why someone might expect there to be more information earlier in the spike trains?
Minor comments:
The authors use the term “conspecific” many times throughout the paper. A quick definition at the beginning of the paper would help the reader.
A number of grammatical errors should be cleaned up (Ex: lines 44, 46, 69, 88, 97, 114, 142, 144, 156, 163, 167)
Lines 73-74: Do the authors mean “smaller”?
Line 97: Do the authors mean p_{Y|X}(y|x)?
The authors should include a key for the blue and red bars in Fig. 2A.
English quality is generally good besides some typos and a number of run-on sentences.
Author Response
We are very pleased that referee 3 also has a positive overall view of the paper; it is difficult in these situation not to feel embarrassed by the number of minor errors they spotted, but we are certainly pleased that the referee took such care. We have addressed all of their concerns: a detailed response is below. We have also attached a "tracked changes" version of the manuscript with new text in blue, substantially changed text in teal and deleted text in red.
We hope that the manuscript is now ready to be accepted, we are certainly pleased with this new version.
All the best
Con
The authors examine the problem of estimating the mutual information between bird songs and spike trains. The authors present a useful set of results that are well motivated. However, I have a number of specific comments and suggestions that I believe can improve the clarity and presentation.
Major comments:
There appears to be a discrepancy between Eq. (5) and Eq. (8). Specifically, should V_i be V in the denominator of Eq. (8), or vice versa? This is important because it guides the entire following discussion.
> thank you, that has been fixed!
I am particularly concerned by the fact that p(y_i) = 1 for all y_i. Should this really be the case?
> this is really the case, any probability mass function has a trivializing change of coordinate, for example, in one dimension changing variables to the cumulative makes the density one. We have added
"and there is always a choice of coordinate that trivializes the density"
>to try to emphasis the confusing point.
It is hard to follow the explanation in the paragraph beginning on Line 96. A clearer explanation would help the reader. In particular, the justification for Eq. (11) is not clear.
> we have substantially rewritten this confusing paragraph, hopefully it is clearer now!
Also, should Eq. (11) read p_{Y|X}(y_i | x_i)?
> yes, thank you, that is fixed now.
I am confused by Eq. (13). If p(y_i) = 1, then the only thing inside the log is p(y_i|x_i), which is given in Eq. (11). But this doesn’t match what’s inside the log in Eq. (13).
> Thanks for spotting this, the mistake was actually in Eq. 11 where we left out an n, it has been fixed now.
At the beginning of the Results, can the authors expand upon why someone might expect there to be more information earlier in the spike trains?
> We have now written in the discussion
"Finally it is observed that it is not the case that the precision of spiking diminishes as the song continues. Since that song can often be identified from the first few spikes of the response, it might be expected that the neuronal firing would become less precise. Precision is metabolically costly. However, although the firing rate falls slightly, the information remains constant on a per-spike basis."
Minor comments:
The authors use the term “conspecific” many times throughout the paper. A quick definition at the beginning of the paper would help the reader.
> "conspecific" means "of the same species"; we realise this is an unnecessary technical term, since we talk often to zebra finch people we'd forgotten it isn't a common term, so we've elimated it by changing "conspecific song" to "zebra finch song".
A number of grammatical errors should be cleaned up (Ex: lines 44, 46, 69, 88, 97, 114, 142, 144, 156, 163, 167)
> thank you for all of these, we believe we have fixed all the ones noted and have worked to eliminate other grammatical errors.
Lines 73-74: Do the authors mean “smaller”?
>we have tried to make this clearer by writing
"with the estimate becoming more-and-more exact for smaller regions $V_i$: if the volume of $V_i$ were reduced towards zero $p_Y(y)$ would be constant in the resulting tiny region."
Line 97: Do the authors mean p_{Y|X}(y|x)?
>yes, thank you: in fact this p_{Y|X}(y|x) no longer appears because of other changes.
The authors should include a key for the blue and red bars in Fig. 2A.
>that has been done.

Round 2
Reviewer 1 Report
All my remarks and comments have been taken into account.
Reviewer 3 Report
The authors have addressed all of my concerns and suggestions.